# Automatic Mapping of Burned Areas Using Landsat 8 Time-Series Images in Google Earth Engine: A Case Study from Iran

**Houri Gholamrezaie** [1], **Mahdi Hasanlou** [1,*], **Meisam Amani** [2] and **S. Mohammad Mirmazloumi** [3]

1   School of Surveying and Geospatial Engineering, College of Engineering, University of Tehran, Tehran 1439957131, Iran
2   WSP Environment and Infrastructure Canada Limited, Ottawa, ON K2E 7L5, Canada
3   Geomatics Research Unit, Centre Tecnològic de Telecomunicacions de Catalunya (CTTC/CERCA), Av. Gauss 7, 08860 Castelldefels, Barcelona, Spain
*   Correspondence: hasanlou@ut.ac.ir

**Abstract:** Due to the natural conditions and inappropriate management responses, large part of plains and forests in Iran have been burned in recent years. Given the increasing availability of open-access satellite images and open-source software packages, we developed a fast and cost-effective remote sensing methodology for characterizing burned areas for the entire country of Iran. We mapped the fire-affected areas using a post-classification supervised method and Landsat 8 time-series images. To this end, the Google Earth Engine (GEE) and Google Colab computing services were used to facilitate the downloading and processing of images as well as allowing for effective implementation of the algorithms. In total, 13 spectral indices were calculated using Landsat 8 images and were added to the nine original bands of Landsat 8. The training polygons of the burned and unburned areas were accurately distinguished based on the information acquired from the Iranian Space Agency (ISA), Sentinel-2 images, and Fire Information for Resource Management System (FIRMS) products. A combination of Genetic Algorithm (GA) and Neural Network (NN) approaches was then implemented to specify 19 optimal features out of the 22 bands. The 19 optimal bands were subsequently applied to two classifiers of NN and Random Forest (RF) in the timespans of 1 January 2019 to 30 December 2020 and of 1 January 2021 to 30 September 2021. The overall classification accuracies of 94% and 96% were obtained for these two classifiers, respectively. The omission and commission errors of both classifiers were also less than 10%, indicating the promising capability of the proposed methodology in detecting the burned areas. To detect the burned areas caused by the wildfire in 2021, the image differencing method was used as well. The resultant models were finally compared to the MODIS fire products over 10 sampled polygons of the burned areas. Overall, the models had a high accuracy in detecting the burned areas in terms of shape and perimeter, which can be further implicated for potential prevention strategies of endangered biodiversity.

**Keywords:** wildfire; remote sensing; mapping burned/unburned areas; Neural Network (NN); Random Forest (RF); Genetic Algorithm (GA); Google Earth Engine (GEE)

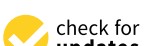



## 1. Introduction

Wildfire is one of the prominent natural disasters that destroys vegetation coverage and increases soil erosion [1,2]. Wildfire also negatively affects global carbon emissions [3,4] and the biodiversity of endangered species [5]. Wildfire may further influence the plant regeneration process and human, wildlife, and microbial activities [6]. Thus, accurate determination of burned areas is crucial to adequately support management plans and adaptation strategies [7–9].

Wildfire has been traditionally detected using field campaigns, which are costly and time-consuming, with limited applicability over spatial and temporal scales [7,10]. Con-

versely, remote sensing approaches combined with advanced image processing and machine learning techniques have become effective alternative tools in distinguishing the areas damaged by forest fire [11,12]. Using satellite observations, one can rapidly and precisely assess the conditions of forests before and after fire events at a relatively lower cost [5,13].

Several global products for mapping burned areas have been developed using satellite data. For instance, the global fire products of Moderate Resolution Imaging Spectroradiometer (MODIS) (MCD64A1) have been available since 2001 at a spatial resolution of 500 m. Most of these ready-to-use products contain a coarse spatial resolution, which results in limited detection of small burned areas [14]. Moreover, the errors of these global products can be observed at many locations [8]. Conversely, the improved spatial and spectral resolutions of several remote sensing missions, such as Landsat and Sentinel, provide a better opportunity to more accurately determine burned areas [15]. Recently, Landsat 8 and Sentinel-2 images have frequently been employed [4,8,12,16–19]. The Landsat time-series is mostly suitable for monitoring spatial and temporal dynamics due to its moderate spatial resolution (30 m), 16-day temporal repeat pass, and the availability of historical data [20].

Many studies have evaluated the effectiveness of developed models using the Landsat 8 and Sentinel-2 missions in detecting burned regions [3,6,10,13]. Initial efforts have compared pre- and post-fire images of Landsat within single scenes to map the changes in vegetation, and understand the extent and severity of burned areas [12,13,21–23]. Further efforts were subsequently focused on wildfire detection using time-series images, along with the spectral bands [24–33], and spectral indices such as Normalized Burn Ratio (NBR), Normalized Burn Ratio Thermal (NBRT), Normalized Difference Vegetation Index (NDVI), Soil Adjusted Vegetation Index (SAVI), and the Enhanced Vegetation Index (EVI) [1,8,24,34,35]. For example, Hawbaker et al. (2017) [8] identified burned areas based on dense time-series data of Landsat and produced the Burned Area Essential Climate Variable (BAECV) for the conterminous United States from 1984 to 2015. However, their algorithm can only be utilized where training data are available. Liua et al. (2018) [1] developed an algorithm based on a harmonic model, and applied to Landsat time-series data acquired from 2000 to 2016 for annual monitoring of the burned areas in southern Burkina Faso. Hong et al. (2018) [36] used a Genetic Algorithm (GA) to obtain the optimal set of wildfire-related variables, and then implemented data mining methods to produce a map of forest-fire susceptibility. Overall, they suggested that the Random Forest (RF) model had better results, and the proposed optimized models outperformed several competing models. Cabrala et al. (2018) [9] investigated the potential of a novel Genetic Programming (GP) method in classifying burned areas. The performance of their approach was investigated using three Landsat images from Brazil (South America), Guinea-Bissau (West Africa), and the Democratic Republic of the Congo (Central Africa), and they concluded that a standard GP method could produce better results compared to the Maximum Likelihood and Classification and Regression Tree (CART) algorithms. Roteta et al. (2019) [4] presented an algorithm to detect small-scale burned areas using Sentinel-2 images and the MODIS active fire product (MCD14ML), whereas the FireCCISFD11 product with a high spatial resolution was consequently generated in Sub-Saharan Africa. Additionally, Roy et al. (2019) [19] mapped burned areas using a combination of Landsat 8 and Sentinel-2A time-series data. They validated a 30 m mapping algorithm of burned areas over $10° \times 10°$ fire-prone regions across Southern Africa. They suggested that precise pre-processing steps were required to reliably combine the data from both sensors. The burned areas were mapped using an RF regression estimator and parameterized with synthetic training data to model the change in reflective wavelength caused by fires. In their work, neither of the Landsat 8 thermal bands were used. Long et al. (2019) [16] also released a product of the Global Annual Burned Area Map in 2015 (GABAM 2015) based on Landsat 8 images. Finally, Hawbaker et al. (2020) [37] refined the BAECV-based algorithm of fire detection. They provided the BAECV products in recent years using atmospherically corrected Landsat 8 Operational Land Imager (OLI) and Thermal Infrared Sensor (TIRS) datasets.

Most products of the burned areas have a coarse spatial resolution and, thus, cannot effectively detect small-scale fire events (e.g., less than 100 ha) [4,16]. Further, apart from a variety of remote sensing indices developed to assess burn severity, as yet there is no consensus about the optimal features used for fire detection [10]. Additionally, feature selection methods or big data processing platforms have not been used in most of the previous algorithms. Therefore, the main objective of this study was to produce a map of burned areas over the entire country of Iran using supervised algorithms, open-access satellite images, and open-source cloud computing platforms. We combined the GA and Neural Network (NN) approaches to select the most optimal features for fire detection. Subsequently, to identify the burned areas, these features were ingested into the NN- and RF-derived models.

## 2. Study Area

This study was conducted in Iran with an approximate area of 1.7 million km$^2$, located in the southwestern part of Asia, with latitudes and longitudes extending from 44° to 64° east and from 24° to 40° north, respectively (Figure 1). Iran is the second-largest biome in Asia, where less than 10% of the country is covered by forests. Owing to the high latitudinal and longitudinal variations, this region includes several biomes, resulting in different ecosystems [38]. The climate of Iran is hot and dry in most regions, with mean annual precipitation and temperature of 250 mm and 17.5 °C, respectively. The rainy season usually occurs from late October through early March, where the maximum rainfall values are recorded during the winter season (Available online: https://www.amar.org.ir (accessed on 20 October 2021)). Owing to these climatic conditions, the forest is the most threatened biome in this country. Fires occur in the dry season, from June to September (Available online: https://www.isa.ir (accessed on 20 October 2021)). Furthermore, Iran is a highly heterogeneous region with seasonally different vegetation types and exposed to the natural and anthropogenic drivers of fire, which makes it more challenging to detect the change in forested and non-forested landscapes [20,39]. This complex nature thus highlights the need for developing an appropriate method of mapping fire-damaged areas over the entire country of Iran for a better interpretation.

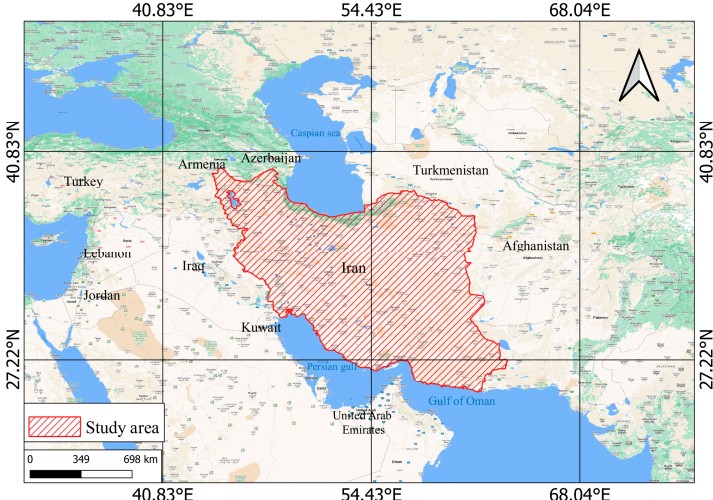

**Figure 1.** The study area: Iran.

## 3. Data

In this study, the surface reflectance time-series images of Landsat 8, which can be accessed in Google Earth Engine (GEE), were utilized. First, the Landsat 8 image collection with 9 bands (B1–B7 and B10–B11) were clipped over the study area for the period of 1 January 2019 to 30 December 2020. Then, 13 spectral indices, shown in Table 1, were generated using the Landsat 8 image and were added to the nine selected bands of Landsat

8. As a result, an image collection with 22 bands was obtained. This process was similarly repeated for the period of 1 January 2021 to 30 September 2021.

**Table 1.** The spectral indices used for wildfire detection in this study.

| References | Equation | Abbreviation | Index Name |
|---|---|---|---|
| [8,40,41] | $1/((0.1 - \text{Red})^2 + (0.06 - \text{NIR})^2)$ | BAI | Burned Area Index |
| [8,40,41] | $(10.0 \times \text{lSWIR}) - (9.8 \times \text{sSWIR}) + 2.0$ | MIRBI | Mid InfraRed Burn Index |
| [8,40,41] | $(\text{NIR} - \text{lSWIR})/(\text{NIR} + \text{lSWIR})$ | NBR | Normalized Burn Ratio |
| [8] | $(\text{sSWIR} - \text{lSWIR})/(\text{sSWIR} + \text{lSWIR})$ | NRB2 | Normalized Burn Ratio 2 |
| [8,41] | $(\text{NIR} - (\text{lSWIR} \times \text{tr1}))/(\text{NIR} + (\text{lSWIR} \times \text{tr1}))$ | NBRT | Normalized Burn Ratio Thermal |
| [8,37,40,41] | $(\text{NIR} - \text{Red})/(\text{NIR} + \text{Red})$ | NDVI | Normalized Difference Vegetation Index |
| [8,40,42] | $(\text{NIR} - \text{sSWIR})/(\text{NIR} + \text{sSWIR})$ | NDMI | Normalized Difference Moisture Index |
| [8,41,43] | $1.5 \times ((\text{NIR} - \text{Red})/(\text{NIR} + \text{Red} + 0.5))$ | SAVI | Soil-Adjusted Vegetation Index |
| [44] | $(\text{NIR} - \text{sSWIR})/(\text{NIR} + \text{sSWIR})$ | NDSWIR | Normalized Difference SWIR |
| [40] | $1/(\text{NIR} - (0.05 \times \text{NIR}))^2 + (\text{lSWIR} - (0.2 \times \text{lSWIR})^2)$ | BAIML | Burned Area Index |
| | | | Modified–LSWIR |
| [40] | $0.2043 \times \text{Blue} + 0.4158 \times \text{Green} + 0.5524 \times \text{Red} + 0.5741 \times \text{NIR} + 0.3124 \times \text{sSWIR} + 0.2303 \times \text{lSWIR}$ | BRI | TassCap Brightness |
| [40] | $-0.1603 \times \text{Blue} - 0.2819 \times \text{Green} - 0.4934 \times \text{Red} + 0.794 \times \text{NIR} - 0.0002 \times \text{sSWIR} - 0.1446 \times \text{lSWIR}$ | GRE | TassCap Greenness |
| [45] | $(\text{NIR}/\text{Green})/(\text{NIR} + \text{Green})$ | GNDVI | Green normalized difference vegetation index |

The coordinates and dates of wildfires that have occurred in Iran were extracted from Iranian Space Agency (ISA) website (Available online: https://www.isa.ir (accessed on 20 October 2021)) for 2020 and 2021. Sentinel-2 images were used to create the peripheral reference data. Moreover, Fire Information for Resource Management System (FIRMS) products were used to assist in creating the non-burned reference polygons, and MODIS (MCD64A1) products were also used to compare and evaluate the output of the models developed within this study, in the GEE environment [46].

## 4. Materials and Methods

### 4.1. Overview

Figure 2 shows a flowchart of the proposed method for detection of burned areas. We used GEE and Colab cloud computing platforms to collect and process time-series images, generate training polygons, export reference data of burned and unburned areas, implement classifications, and validate the resultant accuracies of the models. Details about each step are provided in the following subsections.

### 4.2. Google Earth Engine (GEE) Platform

GEE is an open-source big data processing platform, developed based on a JavaScript programming environment, which provides free of charge access to a variety of earth observations. There is no need to download the datasets when using this service and one can perform different types of analyses and interpretations by collecting the input data and calling the available functions. Furthermore, the analysis and interpretations in this system are feasible with the participation of several processors in a parallel sense, which is much faster than the conventional computational resources [46,47]. This platform has been successfully used in many studies to facilitate mapping of large areas [47–58].

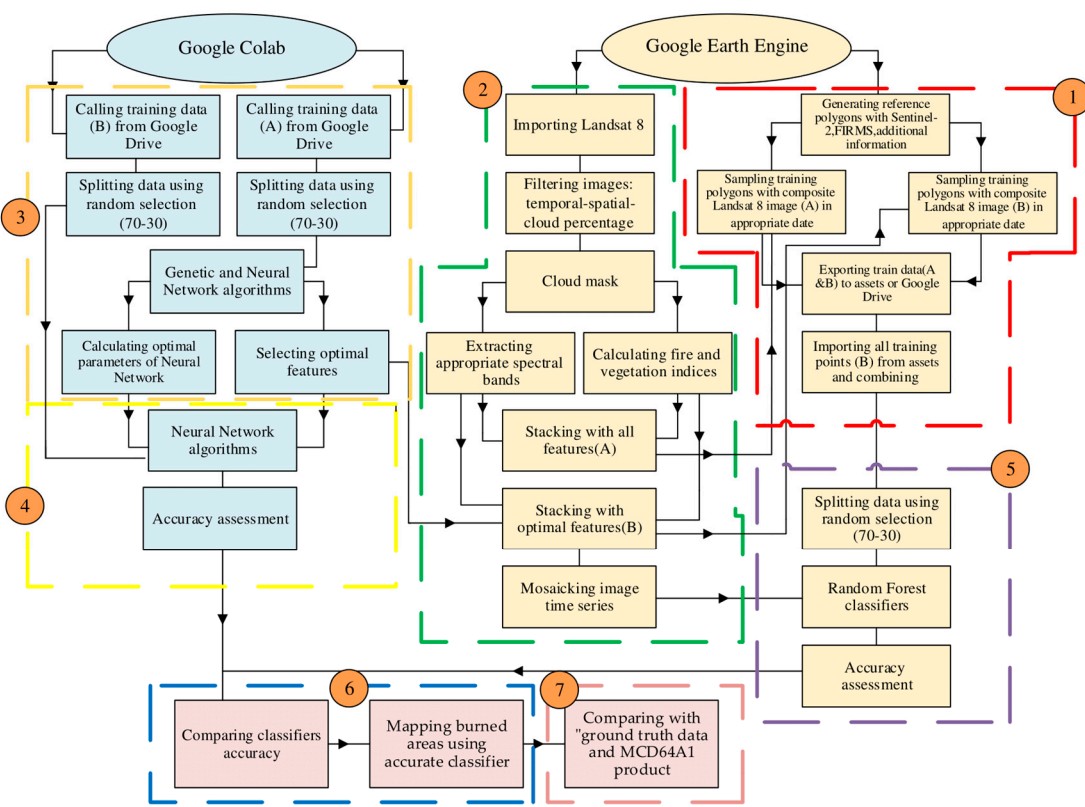

**Figure 2.** Flowchart of the method for identification of burned areas. Generating the reference polygons in (1), extracting further spectral bands of Landsat 8 in (2), selecting optimal bands using GA and NN in (3), assessing the accuracy by the NN in (4) and by RF in (5), evaluating the performance of the RF and NN classifiers in (6), and finally comparing against the ground truth data and MCD64A1 product in (7).

### 4.3. Generating Reference Polygons

Accurate samples of the locations and dates of fire occurrences were considered as the reference data in the derived models. As the main objective of this study was to identify the areas changed by wildfire, samples for the burned and unburned categories were generated using the procedure described in the two subsections below.

#### 4.3.1. Burned Reference Data

The specifications of the fires that occurred in Iran in 2020 and 2021, including the center coordinates and approximate date of the fire events, were extracted from the ISA website (Available online: https://www.isa.ir (accessed on 20 October 2021)). We generated the polygons of the fire areas using the false-color combination of red, near infrared (NIR), and shortwave infrared (SWIR) bands of Sentinel-2, and comparing the images before and after the fires in the GEE platform. By this, 99 polygons of burned areas were created in 2020 (Figure 3), with a total area of 49,935 hectares, which were later used to train and test the models. It is worth noting that the reference data in 2020 were used to train the classifiers in both timespans of 2019–2020 and 2020–2021. Moreover, the polygons of fires that occurred in 2021 were produced as reference data (Figure 3), that is, were not used for training and testing of the models, but were only employed for performance validation of the models and comparison with the fire map obtained in 2021 in the same areas.

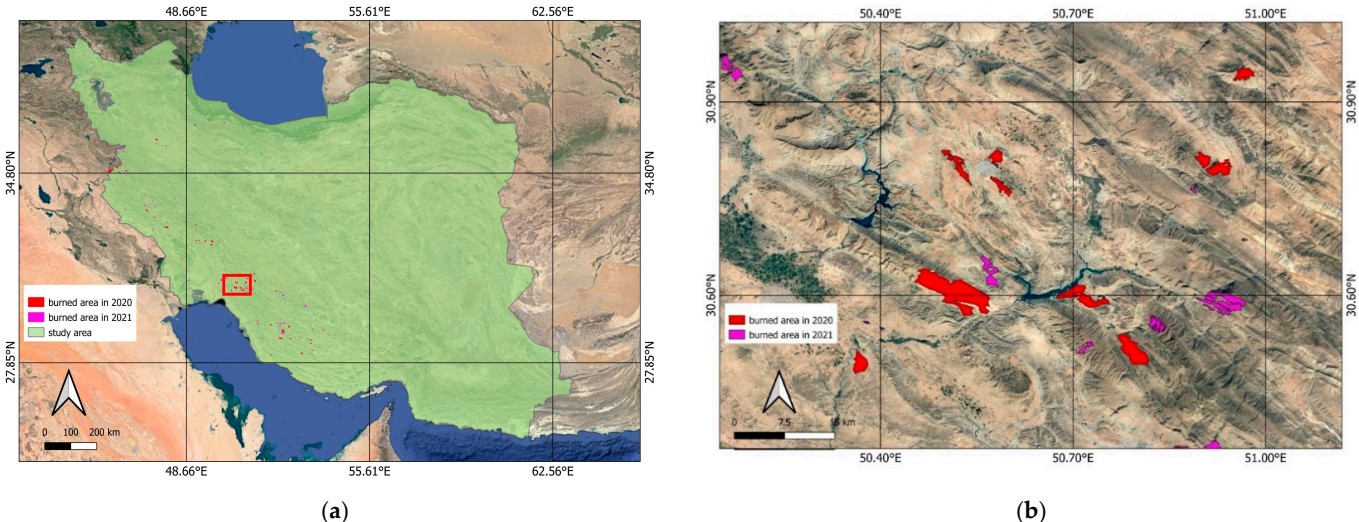

**Figure 3.** Map of (**a**) burned polygons over the study region in 2020 and 2021, with (**b**) a close view of the red box shown in (**a**).

### 4.3.2. Unburned Reference Data

To create reference data for the unburned class, all the burned reference data from the previous step was entered and displayed in the GEE environment. The FIRMS products, which are also available in GEE, were used to detect all the fire alarms at least two years before the target year (2021) to assure the lack of any burned effects in the given period. In the next step, we allocated an arbitrary date to these polygons, and the Sentinel-2 images were also investigated with a color combination of the spectral bands of red, NIR, and SWIR in the periods before and after the arbitrary date to reassure that the above polygons were accurately selected in the unburned areas.

It was assumed that all the alarms caused by the FIRMS products were correct, and we thus created the unburned polygons in the areas that did not belong to either of these two categories. In this step, the unburned data were created from all types of land covers, such as urban areas, rivers, soil, asphalt roads, and mountains. Overall, we created 250 polygons of unburned regions at different sizes (see Figure 4), yet with almost the same area as that of the burned zones (i.e., 49,000 ha) to establish the necessary balance between both types of reference data.

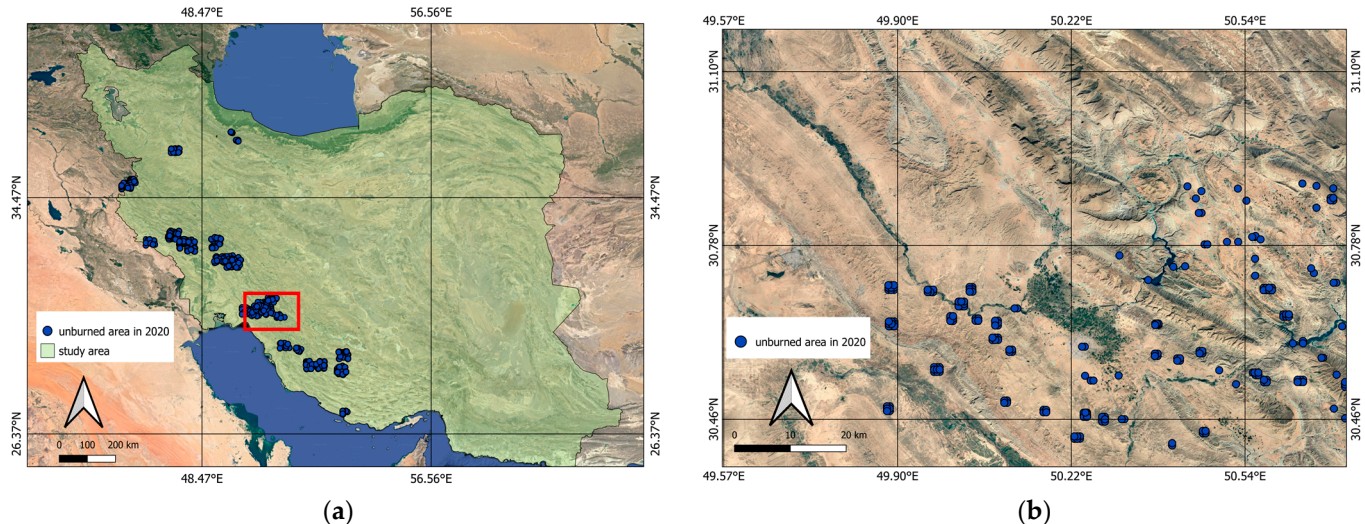

**Figure 4.** Map of (**a**) unburned polygons over the study region in 2020, with (**b**) a close view of the red box shown in (**a**).

### 4.3.3. Preparing Training Polygons

Once the reference polygons were generated, the training dataset was generated using predictor variables. To this end, the *SampleRegions* function in GEE was applied to the polygons on any given image, including spectral bands and indices, to insert the value of predictors into a table and to generate the training samples. The training polygons were thus overlapped by any given image to acquire the predictor variables along with labels [59]. In most studies, the reference data were assumed to be fixed over time and can be resampled with any suitable satellite image, though the burned and unburned areas may vary over time. The samples of burned areas should be thus created regionally using satellite images on the appropriate date after the fire events. Similarly, the samples of unburned areas should be undertaken in the regions that were not exposed to any previous fires. At this stage, we regionally sampled each burned and unburned polygon using the Landsat 8 images in GEE, which are in the form of a data cube containing the bands and indices mentioned in the former subsection (22 features including nine spectral bands of Landsat 8 were included and the 13 indices shown in Table 1), at a 30 m spatial scale.

### 4.4. Mapping Burned Areas

Post-classification is a common approach for analysis of change detection. This method requires two or more datasets to create a change map [51–58]. Post-classification of change was implemented in two stages: (1) each dataset was classified separately, and (2) the results from each classification were compared. To this end, the image differencing (ID) algorithm was used. Finally, a change map was generated [59].

In recent decades, a variety of approaches have been developed for land classification. These include pixel-based (e.g., image classification and regression), subpixel-based, object-oriented algorithms, and Spectral Mixture Analysis (SMA) [2,11,14,28,36,60–65]. In this study, the RF and NN algorithms were used to classify images before and after the date of any given wildfire event [16,18,19,24,66]. For this purpose, the RF algorithm was scripted in the GEE under JavaScript programming environment, and the NN process was encoded in the Google Colab computing platform.

### 4.5. Feature Selection in Google Colab Platform

Optimal spectral indices and bands for burn areas are different from one region to another due to variations in the ecosystems. Therefore, the determination of optimal features is important for producing accurate fire maps on a local scale [6]. Given several limitations in the implementation of feature selection methods in GEE, this step was alternatively implemented in the Google Colab platform. Google Colab is a cloud-based service based on the Jupyter Notebook, providing free of charge access GPU in the runtime [46]. This platform allows the ability to write and run scripts, save and share the analysis of results, and take advantage of powerful computing resources [67]. Implementation of the feature selection algorithms in the Google Colab platform, which are divided into the wrapper and filter methods, can recognize and eliminate unrelated and redundant features and reduce the model complexity. For instance, Stromann et al. (2020) [29], Cai et al. (2018) [68], and Pal and Foody (2010) [69] reported the promising potential of wrapper methods, such as GA [64,70–74], for finding the most-probable solutions without exploring the entire search space [36].

### 4.6. Genetic Algorithm (GA) for Optimal Features Selection

In this study, we implemented a combination of GA and NN for selection of optimal features. A NN architecture has several layers, each of which contains a number of neurons. The first, middle, and last layers of a NN are the input, hidden, and output layers, respectively. The efficiency of NNs depends on several parameters, especially the network architecture, which can conventionally be determined manually based on operator experience or in combination with heuristic solution-search or optimization methods [75].

We determined the optimal features of Landsat images in classification and the most optimal parameters of the NN using the reference data. Accordingly, 70% and 30% of the samples were used as the training and test dataset, respectively. The parameters of the GA library were experimentally set, as shown in Table 2.

**Table 2.** The optimal setting of GA library parameters.

| Parameter | Value |
|---|---|
| max_num_iteration | 10 |
| population_size | 100 |
| mutation_probability | 0.1 |
| elit_ratio | 0.01 |
| crossover_probability | 0.5 |
| parents_portion | 0.9 |
| crossover_type | uniform |
| max_iteration_without_improv | None |

To construct the NN model, we set the loss function equivalent to binary cross-entropy and the metric comparable with accuracy. The activation function was softmax in the last layer. We also determined the optimizer and activation functions for middle layers, the number of layers, and neuron numbers in each layer, while simultaneously specifying the optimal bands (features) using the combined GA and NN algorithm. Overall, we entered ~1,159,995 initial data (in terms of burned and unburned categories) in the Google Colab environment for training the models.

### 4.7. Resampling Reference Polygons

The training polygons of the burned and unburned areas were resampled using Landsat 8 time-series images with the optimal features. According to the results obtained from the previous step, it is not required to use Landsat 8 images with a 30 m sampling scale, as this scale increased the size of the training data and reduced the speed of the calculations without further accuracy improvement. Therefore, the sampling scale of 500 m was considered to resample the training polygons.

### 4.8. Neural Network (NN) Classifier

The NN algorithm was implemented in the Google Colab platform. Based on random selection, the reference samples were split into training (70%) and test (30%) categories. The input specification of the algorithm was defined based on the optimal features, and the parameters of the NN algorithm were specified according to the output from the former step. We subsequently evaluated the accuracy of this model against the test data (see Section 4.10).

### 4.9. Random Forest (RF) Classifier

RF is an ensemble, non-parametric, machine learning classifier [76]. RF is a supervised method comprising several decision trees that individually operate as an ensemble [28]. Each of these decision trees is trained by a subset of training samples (i.e., in-bag samples) and uses the parameters and remnant (i.e., out-of-bag samples) for internal cross-validation. Later, the outcomes are integrated through the bootstrap aggregation technique to combine the classification results from numerous independent random decision trees and to predict the class label [77]. The spectral indices and bands derived from the GA algorithm were ingested into the RF model in GEE. We restricted the number of random trees to 500 to balance between efficiency and accuracy of the classification models.

The RF algorithm is optimized according to the number of regression trees (ntree) and the number of predictors (mtry) at each split as it creates a new tree (node). The mtry predictor at each node affects the accuracy of a tree and increasing the ntree number increases the performance of models. To determine the optimized values of mtry and

ntrees, we used repeated thrice from a range of 1–6 mtry and 50–1500 ntrees. The model with the highest accuracy was consequently selected as the final model [3]. At this step, the areas affected by wildfire were identified using the RF classification in the two timespans. Specifically, as we aimed to identify the changed areas in 2021, the first classification was formed from 1 January 2019 to 30 December 2020 and the second classification was performed from 1 January 2021 to 30 September 2021.

*4.10. Accuracy Assessment and Performance Comparison*

We used the accuracy indices extracted from the confusion matrix of the classification to evaluate the performance of the RF and NN algorithms [78–80]. These indices included Overall Accuracy (OA), Kappa index (K), User Accuracy (UA), and Producer Accuracy (PA). As shown in Figure 5, we also randomly selected 10 polygons of the burned areas in 2021 to evaluate the performance of the models. The output of the models was visually compared with the polygons of burned areas as the ground truth as well as with the associating MODIS products (MCD64A1), in the GEE platform.

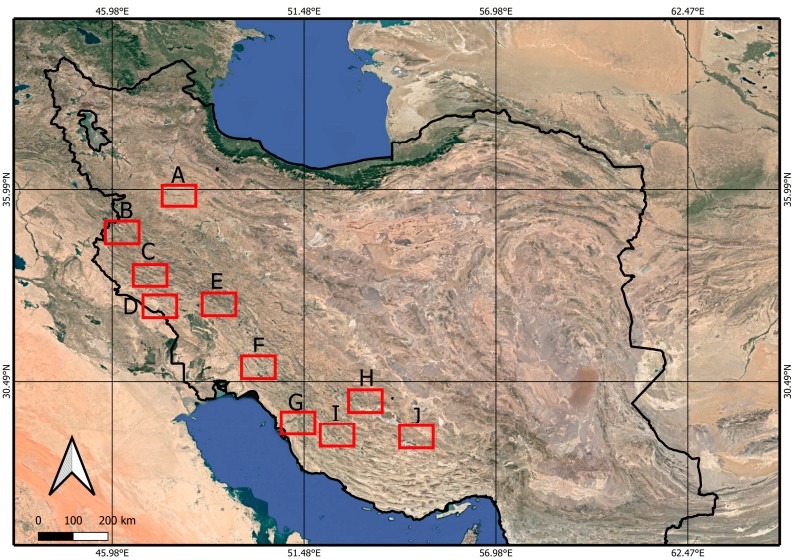

**Figure 5.** Randomly sampled polygons of burned areas in Iran are delimited with red boxes as the ground truth to evaluate the performance of the output of derived models.

*4.11. AccMODIS Direct Broadcast Burned Area Collection 6 (MCD64A1)*

The MCD64A1 product has been released to identify the changes in wildfire-affected lands over large regions since 2001, thanks to the thermal infrared bands of onboard sensors. To provide this latest product, a multi-step approach was implemented to map the burned areas using MODIS images (500 m) and active fire locations (1 km). At first, daily time series of surface reflectance was generated for each pixel from MODIS bands 1 (0.620–0.670 μm), 5 (1.230–1.250 μm) and 7 (2.105–2.155 μm). A daily vegetation index sensitive to burn was then computed from the MODIS shortwave bands 1240 and 2130 nm (VI = (B5 − B7)/(B5 + B7)). The sudden change in this index was considered as the indication for detection of the burn-driven spectral changes. The surface reflectance time series was analyzed to derive the statistical measures for temporal variability based on the dynamic thresholds. Finally, and the active fire locations were used to provide probability density functions for filtering the training dataset and the subsequent determination of burned and unburned pixels with spatial resolution of 500 m [81–83].

In recent years, in studies like ours where burned areas have been detected using medium-resolution images such as Landsat-8 or Sentinel-2, the MCD64A1 product was used to evaluate the output of algorithms [2,14,19,24,37]. We thus used this product, as

available in the GEE data catalog, delimited over the country of Iran in 2021 to assess and compare with the output of our algorithm.

## 5. Results

### 5.1. Combination of Genetic Algorithm (GA) and Neural Network (NN) Classifiers

Optimum features were selected after the combined implementation of the GA and NN algorithms. Due to the high number of training points, high accuracy was obtained by the NN (Figure 6). As shown in Table 3, 19 out of 22 features were selected using GA. This algorithm also determined the optimal number of layers and neurons in each layer, as well as the activation function of the hidden layers and optimizer as listed in Table 4. The architecture of NN classifier was thus constructed as schematically shown in Figure 7.

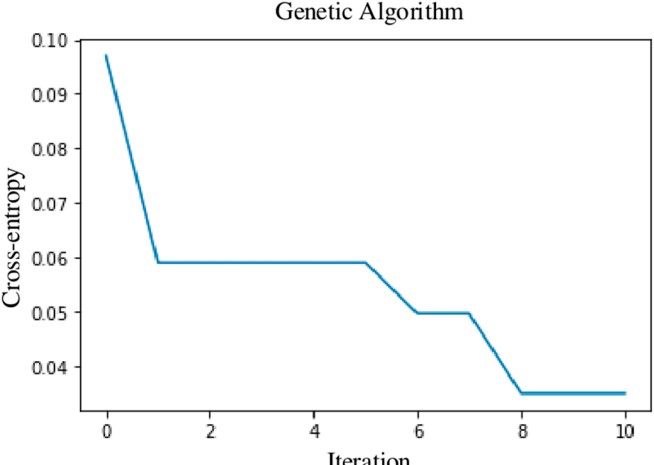

**Figure 6.** Relationship between the loss function and the number of iterations in implementation of the GA.

**Table 3.** The optimal features selected by the GA.

| Optimal Indices | Optimal Bands |
|:---:|:---:|
| NBRT | Red |
| NBR2 | Green |
| NBR | Unblue |
| MIRBI | Blue |
| NDVI | NIR |
| NDMI | SWIR1 (sSWIR) |
| NDSWIR | SWIR2 (lSWIR) |
| GNDVI | Thermal Infrared 1 (tr1) |
| SAVI | Thermal Infrared 2 (tr2) |
| BAI | |

**Table 4.** Optimal parameters of NN recognized from the GA.

| Optimizer | Adam |
|:---:|:---:|
| activation | relu |
| nLayers | 7 |
| nNeurons | [13,41,49,52,54,57,79] |

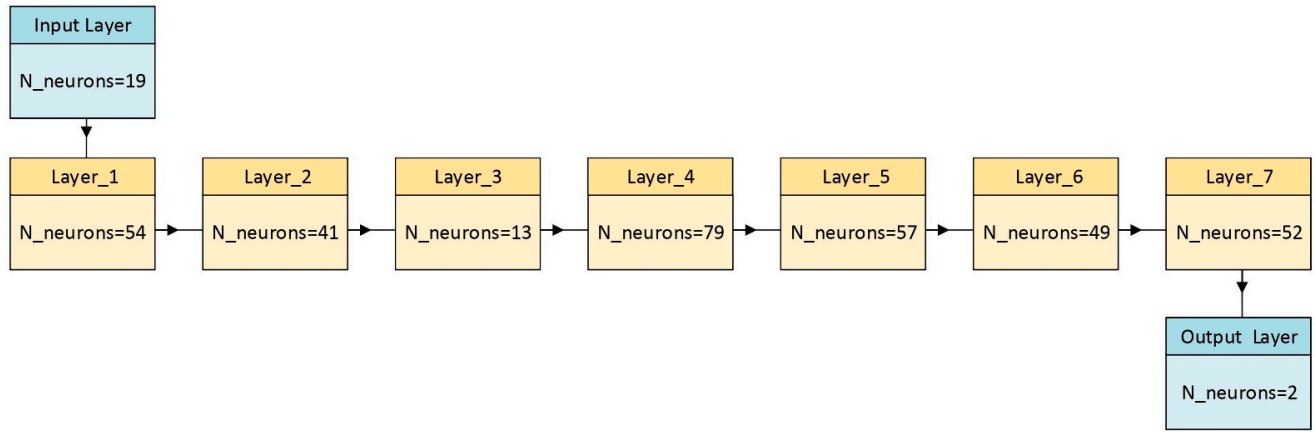

**Figure 7.** Architecture of NN framework resulting from the combination of GA and NN approach.

After selecting the optimal features, the training polygons were consequently adopted from Landsat images. At this stage, we applied the *SampleRegions* function on the time-series with a scale of 500 m to cope with the GEE limitations in case of a high number of training points. As such, 4812 training points were created within the burned and unburned areas. The NN algorithm was again implemented with the above-mentioned parameters and using the new training data based on the architecture shown in Figure 7. Overall, 15,095 parameters were trained in this network and the evaluation criteria in terms of confusion matrix were obtained as listed in Table 5. The confusion matrix inferred from the implementation of this model is shown in Figure 8. This matrix showed that 707 and 650 points were correctly categorized in the unburned and burned classes, respectively. In contrast, 54 points were incorrectly allocated to the burned class, while 33 points were wrongly classified to the unburned class.

**Table 5.** Confusion matrix of the NN classifier.

| Total of Test Data = 1444 | Unburned Class | Burned Class | NET | User Accuracy (%) | Commission Error (%) |
|---|---|---|---|---|---|
| Unburned class | 707 | 54 | 761 | 93 | 7 |
| Burned class | 33 | 650 | 683 | 95 | 5 |
| Total | 783 | 661 | | Overall accuracy (%) = 94 | |
| Production accuracy (%) | 90 | 98 | | Kappa coefficient (%) = 88 | |
| Omission error (%) | 10 | 2 | | | |

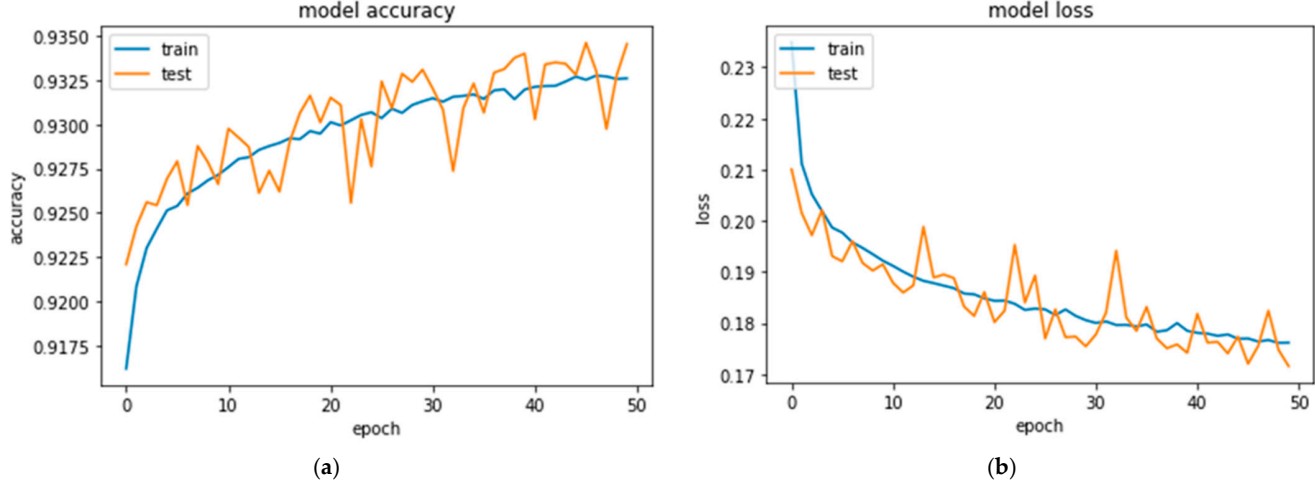

(**a**)　　　　　　　　　　　　　　　　　　　(**b**)

**Figure 8.** Comparison of (**a**) accuracy and (**b**) loss functions of the NN algorithm derived from the training and test datasets.

Additionally, based on the results illustrated in Figure 8, the performance of the resultant model was coherent between both training and test data, indicating the appropriate setting of parameters and suitable construction of NN architecture.

### 5.2. Random Forest (RF) Classifier

We applied the RF algorithm to the Landsat images and obtained the confusion matrix and the corresponding accuracy measures, where the results are provided in Tables 6 and 7.

**Table 6.** Confusion matrix of the RF classifier.

| Total of Test Data = 1444 | Unburned Class | Burned Class | NET | User Accuracy (%) | Commission Error (%) |
|---|---|---|---|---|---|
| Unburned class | 742 | 19 | 761 | 98 | 2 |
| Burned class | 41 | 642 | 683 | 94 | 6 |
| Total | 783 | 661 | | Overall accuracy (%) = 96 | |
| Production accuracy (%) | 95 | 97 | | Kappa coefficient (%) = 90 | |
| Omission error (%) | 5 | 3 | | | |

**Table 7.** Different accuracy measures of the RF classifier derived from the confusion matrix.

| ACC = TP + TN/(TP + FN + FP + TN) | Overall accuracy | 0.96 |
|---|---|---|
| TPR = TP/(TP + FN) | Sensitivity or recall | 0.97 |
| FPR = FP/(FP + TN) | Probability of false alarm | 0.06 |
| TNR = TN/(TN + FP) | Specificity | 0.94 |
| FNR = FN/(TP + FN) | Miss rate | 0.02 |
| PPV = TP/(TP + FP) | Precision | 0.95 |
| NPV = TN/(TN + FN) | Negative predictive value | 0.97 |
| FOR = FN/(FN + TN) | False omission rate | 0.03 |
| FDR = FP/(TP + FP) | False discovery rate | 0.05 |

The results revealed that 742 and 642 points were correctly classified in the unburned and burned classes, respectively. Moreover, 19 and 41 points were incorrectly assigned to the burned and unburned classes, respectively. According to Figure 9, the NBRT, NBR2, and NBR features had the highest importance in the RF classifier. In general, the RF had higher accuracy compared to the NN algorithm (compare Tables 5 and 6). However, both algorithms were subject to the commission errors in identifying the burned and unburned areas, potentially due to several factors such as the burned-like lands (i.e., dry, senesced, or dead vegetation as well as harvested agricultural lands) and different types of land cover classes with similar spectral response.

Figure 10 shows the output derived from the post-classification method and RF algorithm. Visual inspections showed that our model accurately detected several wildfire-affected areas that were not included in the ISA website, which indicated the superiority of the proposed approach. As an alternative hypothesis, these inconsistencies were partially due to the commission errors, which were mainly due to the difficulties in discriminating between the spectrally similar land covers. The inclusion of vegetation type maps and ISA information could contribute to mitigating the separability errors and improving the overall classification accuracy.

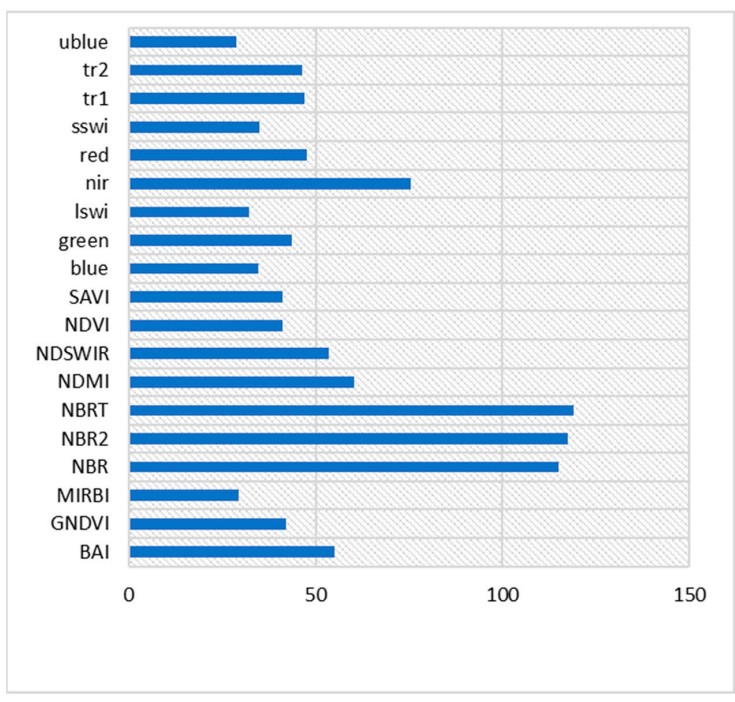

**Figure 9.** Graph showing the importance of input variables derived from the RF algorithm in GEE.

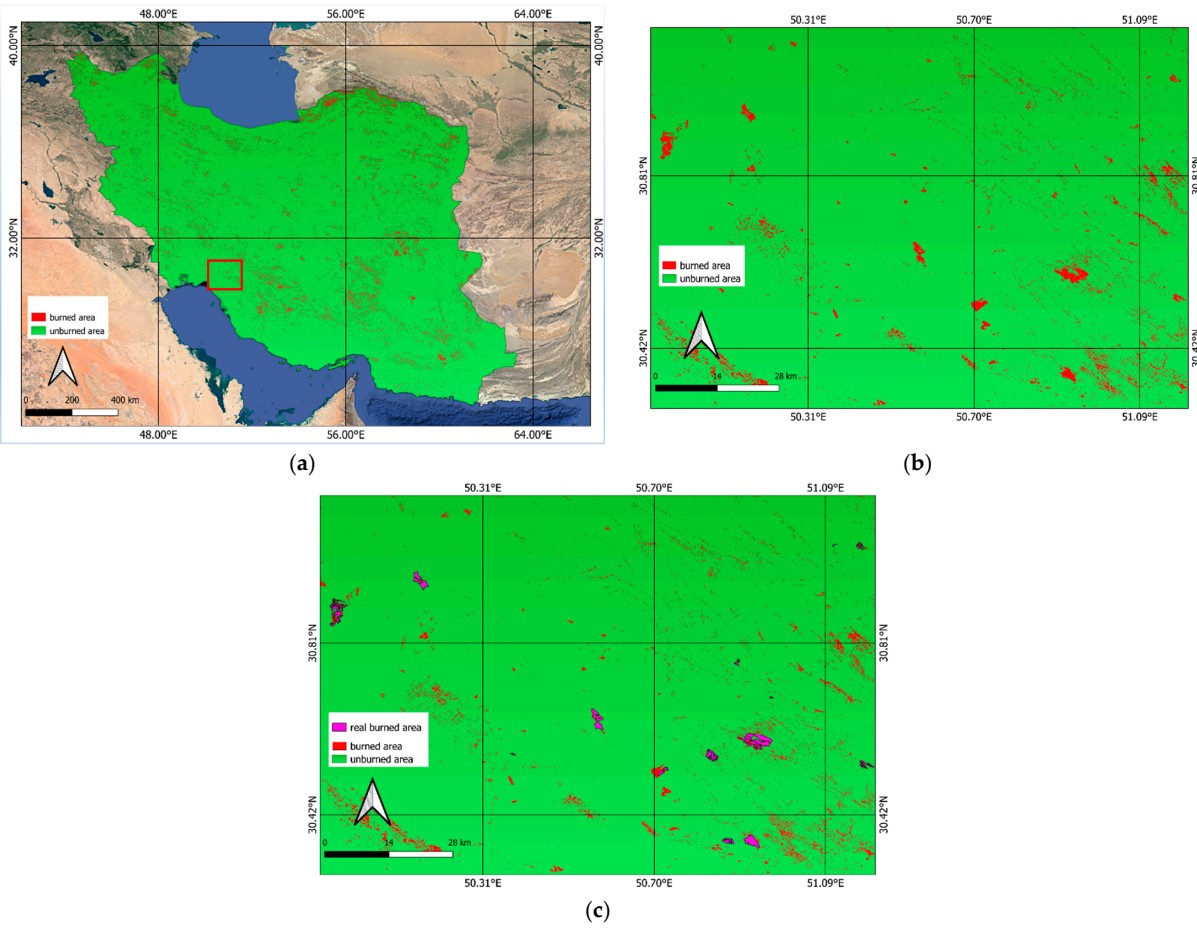

**Figure 10.** Map of (**a**) identification of change in areas affected by wildfire in 2021. Zoomed views (**b**,**c**) of the southwestern regions delimited by red box in (**a**).

*5.3. Validation*

The output of the RF model was compared with the generated polygon of fires in 2021 (as mentioned above) as the ground truth data and MODIS products over the 10 selected areas (Figure 11). The output shows that the proposed RF model correctly identified most of the burned areas. As is clear in Figure 11B–E,G–J, there are many pixels which were identified as "burned" by the proposed model, possibly due to the spectrally similar land cover types. Furthermore, the MODIS products have limitations in identifying the burned regions in Figure 11A,C–E,J mainly due to the coarse spatial resolution. On the other hand, our approach mostly distinguished these relatively small, burned areas, emphasizing the promising potential of our approach. Overall, the results showed that the proposed model could efficiently identify the burned areas.

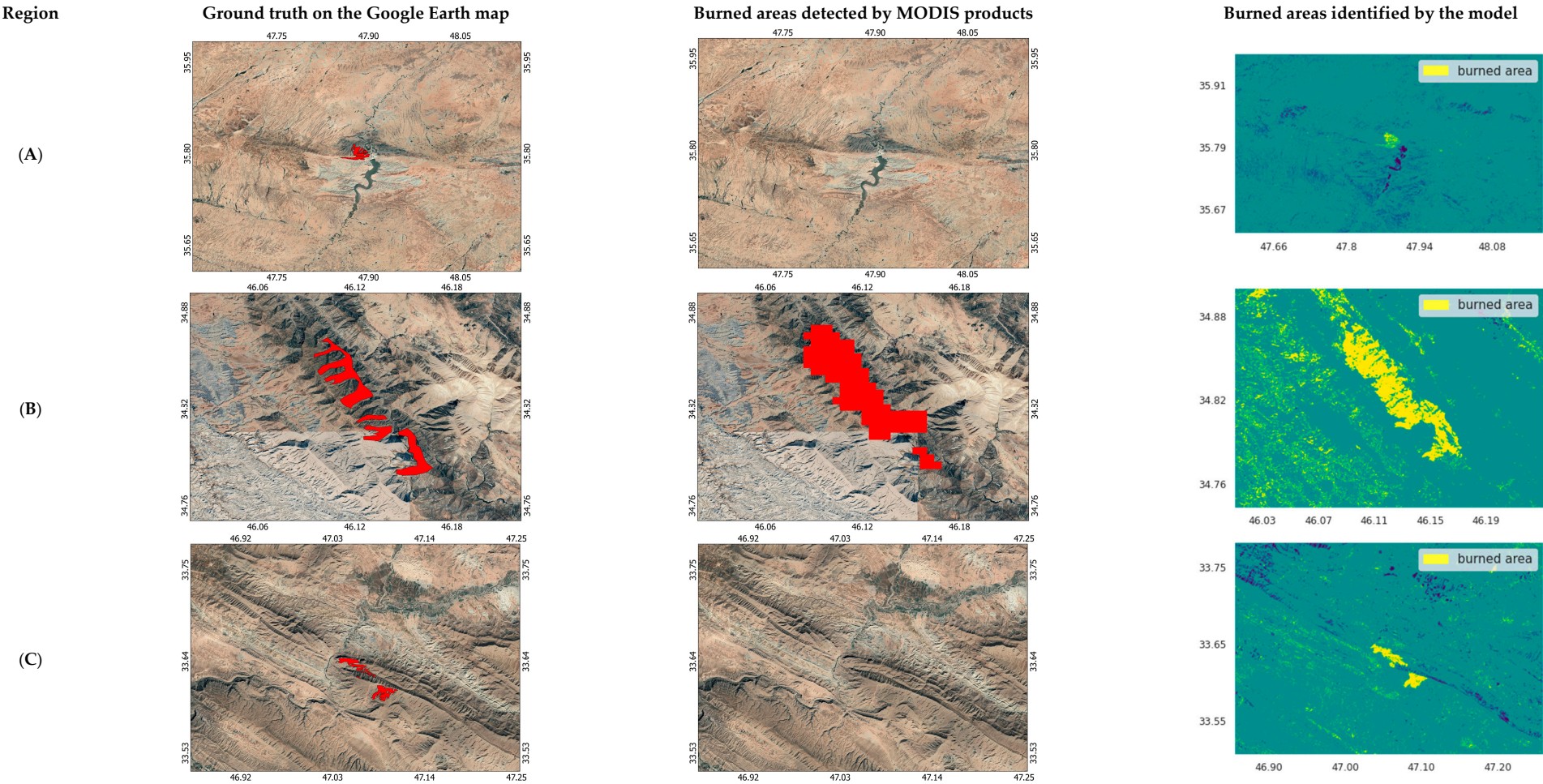

**Figure 11.** *Cont*.

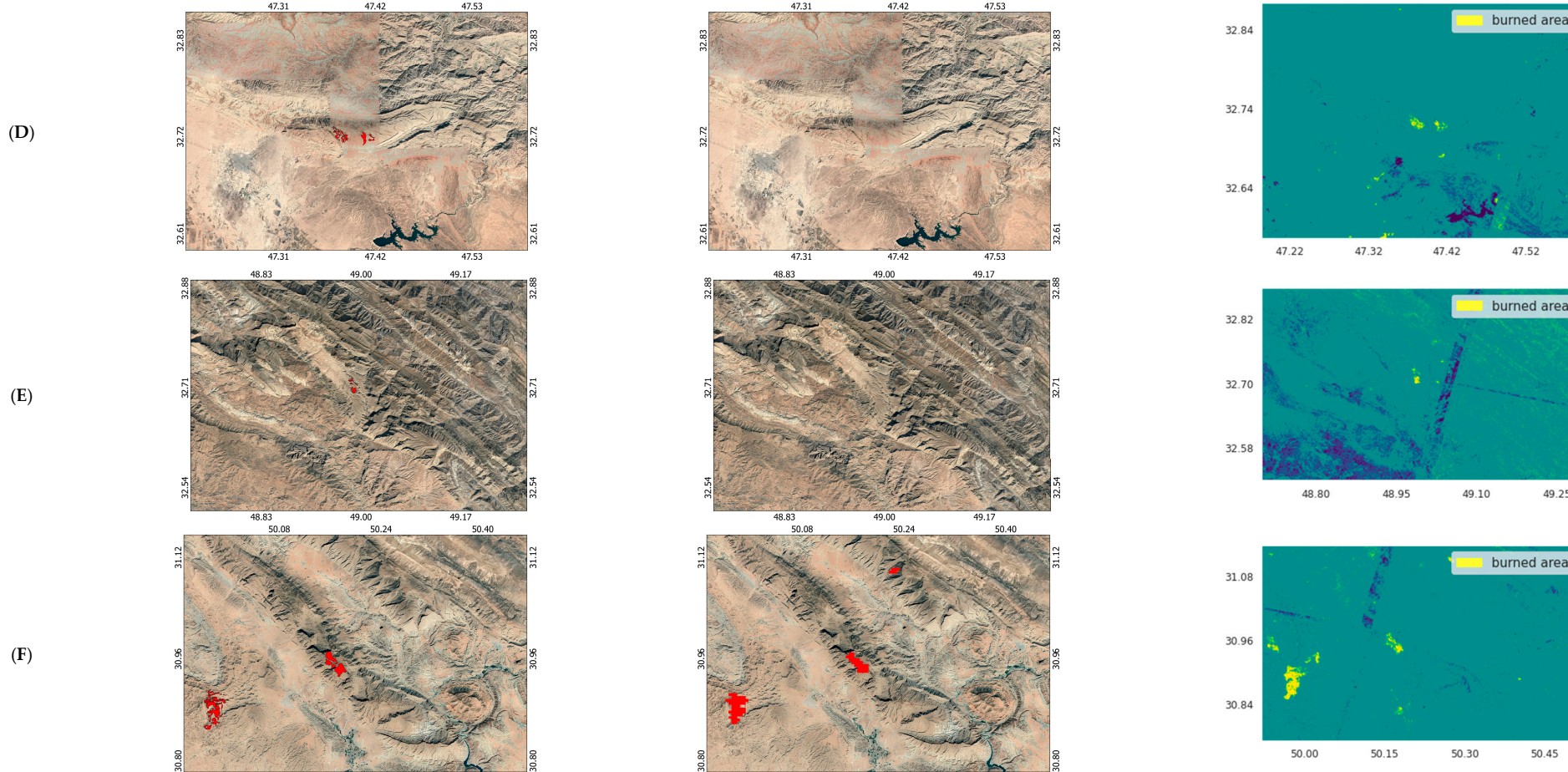

(**D**)

(**E**)

(**F**)

**Figure 11.** *Cont.*

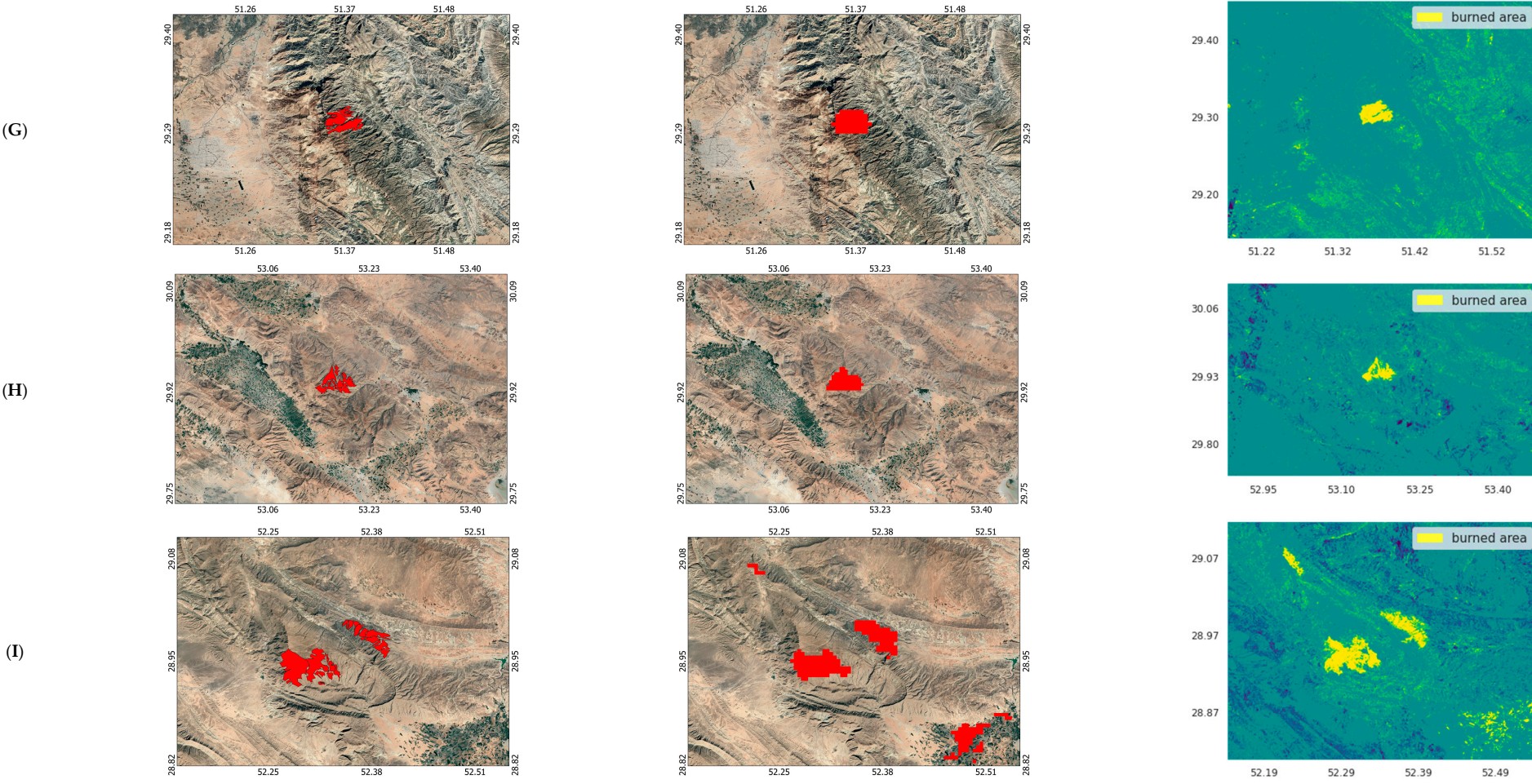

**Figure 11.** *Cont*.

(J)

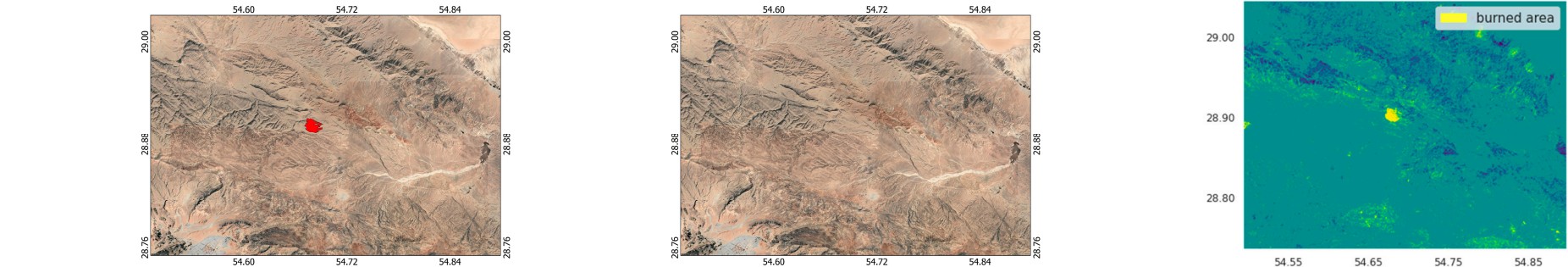

**Figure 11.** The left, middle, and right columns respectively show the actual burned areas in red reported by the Iranian Space Agency as the ground truth overlayed on the Google Earth map, the burned areas in red detected by MODIS products overlayed on the Google Earth map, and the burned areas in yellow identified by the proposed model in different regions (**A–J**).

## 6. Discussion

In this research, we have taken advantage of the capabilities of both GEE and Google Colab computing platforms, such that this two-way relationship could attempt to overcome the limitations of using each one alone. As shown in previous studies [46,47] application of the GEE resources to large study regions is of importance, as there is no need to download the images and allocate storage space. Further, the implementation and computations are facilitated by fast and easy access to a variety of data catalogs and ready-made products, the availability of machine learning algorithms such as RF, and the potential for simple scripting in JavaScript environment. Given the large size of Iran as our study area, resorting to this cloud computing space was logical and cost-effective. As the aim of this study was to identify small-scale fires and ready-to-use fire products are generally based on low-resolution MODIS images, we decided to use images with a higher spatial resolution. Considering the success of previous studies in detection of wildfire using Landsat 8 images, the global annual burned area map (GABAM) products and other regional fire products [1,8,16,19,37], in this study we likewise used Landsat 8 images. We also used the popular image differencing and classification methods to identify the changes in areas caused by wildfire, and further spectral indices were amended to the Landsat spectral bands. Our work goes beyond the previous studies [1,8,24,34,35] to produce multiple spectral indices of vegetation and fire (13 indices), and subsequently we considered Landsat images with the 22 bands. This investigation showed that GEE has a high potential for feature extraction. According to previous studies, to achieve higher accuracy the images introduced to the classifiers must have optimal features. An optimization algorithm is also required. To do so, the combination of the GA and the NN approaches have mostly been used [67,70–74]. Since implementation of these algorithms is easier using the strong libraries of Python and because the training data was mandatory for training the models, we thus migrated to the Google Colab environment.

As the coordinates and the dates of recent fires in Iran are disseminated through the ISA website, we produced the final polygons of burned and unburned areas using this information, Sentinel-2 images, and FIRMS products, with approximately the same total area in order to balance between both types of datasets. We then sampled each of these polygons on a suitable date using Landsat 8 images with 22 bands, and sent to the Google Drive with a spatial scale of 30 m. We repeated this process for a large number of polygons separately, in which the optimal features were finally obtained. Consequently, we implemented the NN and RF algorithms in Colab and GEE environments respectively, to compare the performance accuracy. Notably, to implement the NN and RF algorithms, it was necessary to re-sample the training data with Landsat 8 images with the number of optimal features. The initial tests, however, reveal that the use of spatial scale of 30 m in this step caused computation errors beyond the memory capacity, and also decreased the speed of the engine in Colab. For this reason, the images were inevitably sampled again with a spatial scale of 500 m to create the training data. This study thus showed that due to the dynamic nature of fires, creating training data is one of the most time-consuming tasks in this kind of research. Further, this comparison showed that despite the higher complexity and more difficult implementation of the NN, the RF algorithm yielded higher accuracy. Finally, in order to evaluate the efficiency of the performance, the results of this research were compared with the polygons produced in the target year (2021) as the ground truth data as well as the MODIS products (MCD64A1) which are available in GEE. As is clear in Figure 11A,C,D,E,J, the methodology succeeded in detecting wildfires in areas that were not detected by the MODIS products. The detected fires in terms of shape and perimeter were similar to the ground truth data in many parts.

One of the limitations of this research is the uncertainty of identifying and entering all recent fires in Iran on the ISA website. Though this does not cause a serious problem in training of the models, it adversely affects evaluation of the results. For instance, if a fire was detected by the algorithm in the target year that is not present in the ground truth

data, it cannot be concluded that this was the commission error of the algorithm, as it may instead be a result of ground truth incompleteness.

Although we intended to create the polygons of all the fires, running the RF algorithm with all the reference polygons was beyond the computational memory of engine or GEE, so we re-sampled the images with a spatial scale of 500 m as the only way to overcome this issue. We found that the extent of the training polygons is not important, but they should be produced in many and different areas. As most of the fires occurred in the west of Iran, the unburned areas were designed in the vicinity of these region to realize the same environmental conditions. As such, the reference data was not defined in the east of the country.

As another limitation of this research, since the goal was to identify small-scale burned areas, it would have been better to use the fire products on the same scale to evaluate the output of the algorithms. However, this was not yet possible as these types of products for the region of Iran have so far been unavailable in the GEE platform. This study showed that the temporal resolution of Landsat images over Iran is not solely enough to identify the burned areas due to the weather conditions in some parts, and as such, the short-term active fires may not be detected until the access to appropriate images is feasible. To reduce the observational gap, it is furthermore recommended to use other available images such as Sentinel-1 and -2. Owing to the large biological difference in the different regions of Iran, it is better to separately develop and train each part of the study area that has homogeneous conditions, and finally combine the output of different parts together.

## 7. Conclusions

The quantitative and qualitative determination of post-fire effects in forests is important for understanding the response of ecosystems to human-made and natural disasters over diverse spatial and temporal scales. Due to the high rate of wildfire occurrences throughout different parts of Iran in recent years, it is mandatory to develop an appropriate approach to detect the affected regions. In this study, reliable methodology was proposed to map the burned areas over the entire country using a post-classification supervised technique applied to the Landsat 8 time-series images. Our proposed methods were implemented in the GEE and Google Colab platforms, which are powerful, free, and computationally fast to monitor and identify fires across large areas. This study showed that the production of reference data is one of the most time-consuming, yet important tasks in discriminating the burned areas. This is basically inherited from the dynamics of the burned areas, which reduces the extent of the burned areas over time due to fire extinguishing or vegetation recovery. Furthermore, an area defined as an unburned class may have experienced another wildfire in former times. First, within the GEE environment, the training polygons of burned areas were created using information from the ISA website and Sentinel-2 images, and the polygons of unburned areas were created through FIRMS fire products in areas with different usages. Then, these polygons were created using Landsat 8 images comprised of 22 features and were sampled at a 30 m spatial scale and converted into reference points and introduced into the combined algorithm of GA and NN for selection of the optimal features and training the models. After selecting 19 optimal features, training polygons were sampled and turned into the reference points using the Landsat 8 images with these optimal features and a spatial scale of 500 m, in GEE platform. Subsequently, these reference points were used as the training data for burned area detection using the NN and RF algorithms. It was observed that these models had low rates of omission and commission errors, noting that the commission errors were mainly due to the spectral similarity of burned areas to harvested lands in agriculture. Moreover, the accuracy of NN performance was primarily affected by several factors, though the network architecture was the dominant driver. The architecture was indeed determined within an empirical process, which was time-consuming specifically for large-scale tasks of classification. We overcame this limitation by implementing the GA and achieved a high accuracy.

**Author Contributions:** Conceptualization, H.G., M.H. and M.A.; Methodology, H.G. and M.H.; Validation, H.G.; Writing—original draft, H.G.; Writing—review & editing, H.G., M.H., M.A. and S.M.M.; Supervision, M.H. and M.A. All authors have read and agreed to the published version of the manuscript.

**Funding:** This research received no external funding.

**Data Availability Statement:** The coordinates and dates of wildfires occurred in Iran in 2020 and 2021 has been extracted from Iranian Space Agency website. Google Earth Engine has been considered as an appropriate platform to use the Landsat 8 images as the main dataset, and the Sentinel-2 images has been used for creating the reference data. Also, Fire Information for Resource Management System and MODIS (MOD14A1) products have been used in the Google Earth Engine environment.

**Conflicts of Interest:** The authors declare no conflict of interest.

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
