# Peer review of "Automatic Mapping of Burned Areas Using Landsat 8 Time-Series Images in Google Earth Engine: A Case Study from Iran"

_remotesensing, doi:10.3390/rs14246376_

Round 1
Reviewer 1 Report
The authors proposed a process to produce a map of burned areas over the entire country of Iran using supervised algorithms, open-access satellite images, and open-source cloud computing platforms. The paper's general concept was well-planned, and the conclusion is convincing. I believe that this manuscript should be published after a suitable revision.
The important part; There is no discussion section, so it should be added. The results of the study should be compared with the findings of the relevant studies published in recent years to support the contributions and novelties of the study.
Unburned areas on figure 4 are not clearly visible.
in validation section, details of MODIS data product used should be added.
The maps given in figure 11 should be made better visual.
Reviewer 2 Report
I think it would be reasonable in terms of validation and fire sites' classification accuracy assessment to use high resolution scenes (> 1 m pixel, as well as high resolution aerial scenes), comparing with Landsat 8, MODIS, and Sentinel-2 data interpretation. This might be reasonable not only for fire areas and boundaries detection, but also for the types of forest vegetation and amounts of burning materials, as plant fuels in the burned areas.
Author Response
Reviewer #2: I think it would be reasonable in terms of validation and fire sites' classification accuracy assessment to use high resolution scenes (> 1 m pixel, as well as high resolution aerial scenes), comparing with Landsat 8, MODIS, and Sentinel-2 data interpretation. This might be reasonable not only for fire areas and boundaries detection, but also for the types of forest vegetation and amounts of burning materials, as plant fuels in the burned areas.
Response: We appreciate the comprehensive review by the Reviewer #2. We did our best to address your kind suggestion which has helped us to improve the quality of the manuscript and thinking for ongoing research in this interesting field.
The validation is now performed with the new MODIS products of MCD64A1, substituting the previous products with a resolution of 1 km with this new products with a resolution of 500 m (please see Section 4.11 on Page 11). We further highlight that yet there is no other product available in Iran for the comparison purposes, which makes an open door for ongoing research in the future.